# Stochastic Deep Networks with Linear Competing Units for Model-Agnostic Meta-Learning

## Abstract

This work addresses meta-learning (ML) by considering deep networks with stochastic local winner-takes-all (LWTA) activations. This type of network units result in sparse representations from each model layer, as the units are organized into blocks where only one unit generates a non-zero output. The main operating principle of the introduced units lies on stochastic principles, as the network performs posterior sampling over competing units to select the winner. Therefore, the proposed networks are explicitly designed to extract input data representations of sparse stochastic nature, as opposed to the currently standard deterministic representation paradigm. We posit that these modeling principles, inspired from Bayesian statistics, yield representations of stronger generalization capacity; this is of immense importance in the case of ML, which is the focus of this work. As we experimentally show, our approach produces state-of-the-art predictive accuracy on standard few-shot image classification benchmarks. This improvement comes with an immensely reduced network sized required for achieving this accuracy; this amounts to a parameter reduction by *one order of magnitude* on average compared to the current state-of-the-art.

## 1 Introduction

When we train machine learning models on problems with limited amounts of training data, we cannot usually get good predictive performance (Lai, 2019; Sculley et al., 2015). This comes in contrast to the human ability to quickly derive information from a range of different tasks, and then adapt to a new task with limited available new examples (Kühl et al., 2020; Castro et al., 2008). In essence, this capability of the mind of learning how to learn (Black et al., 2006) has inspired researchers to investigate the concept of Meta-Learning (ML) (Lake et al., 2017; Vilalta & Drissi, 2002; Wang et al., 2016; Zoph & Le, 2017; Flennerhag et al., 2020; Lake et al., 2015; Hutsebaut-Buysse et al., 2019).

There is a large variety of deep learning methods for ML (Finn et al., 2019; Baik et al., 2020; Andrychowicz et al., 2016). Specifically, Finn et al. (2017) presented the *Model-Agnostic Meta-Learning* (MAML) algorithm that enables tuning the parameters of a trained network to quickly learn a new task with only a few gradient updates. To get away with the entailed second-order computations, which are expensive, several researchers proposed appropriate first-order approximations for MAML; such works are the *First-Order MAML* (FOMAML) of Finn et al. (2017) and the *Reptile* algorithm of Nichol et al. (2018). Recently, several researchers have also considered Bayesian inference-driven methods for deep learning ML, extending upon older works built for conventional machine learning approaches (Yoon et al., 2018; Finn et al., 2018; Ravi & Beatson, 2019; Patacchiola et al., 2020; Zou & Lu, 2020; Grant et al., 2018; Chen et al., 2020).

This work proposes a different regard toward improving generalization capacity for deep networks in the context of ML. Specifically, our proposed approach relies on the following main concepts:

- **The concept of sparse learned representations.** For the first time in the literature of deep network-driven ML, we employ a mechanism that inherently learns to extract sparse data representations. This consists in replacing standard unit nonlinearities (e.g., ReLU)

with a unit competition mechanism. Specifically, (linear) units are organized into blocks. Presented with some input, the units within a block engage in a competition process with only one winner. The outputs of all units except for the winner are zeroed out; the output of the winner retains its computed value (local winner takes-all, LWTA, architecture).

- **The concept of stochastic representations.** We establish a stochastic formulation for the previously described competition process. Specifically, we postulate that, within a block of competing units, winner is selected via sampling from an appropriate Categorical posterior. The corresponding winning probability of each unit is proportional to its linear computation (thus depending on the layer input). Via this competition process, we yield stochastic representations from network layers, that is representations that may change each time we present to the network layer exactly the same input.

Based on the results from existing approaches, we posit that the proposed treatment of the ML problem, which combines learned representation sparsity and stochasticity, will be extremely beneficial to the deep learning community. We dub our approach Stochastic LWTA for ML (StochLWTA-ML).

We perform a variational Bayes treatment of the proposed model. We opt for a full Bayesian treatment, by also handling network weights as latent variables. That is, we elect to impose an appropriate prior over the network weights and fit approximate (variational) posteriors. We evaluate our approach on a number of standard benchmarks in the field, namely Omniglot (Lake et al., 2017), Mini-Imagenet (Vinyals et al., 2016) and CIFAR-100 (Krizhevsky, 2009). We show that our approach offers a variety of advantages over the current state-of-the-art methods, namely: (i) incurring reduced predictive error rate compared to the currently state-of-the-art methods in the field; (ii) obtaining this performance with networks that comprise *one order of magnitude less* trainable parameters, and therefore give rise to better computational efficiency and imposed memory footprint.

The remainder of this paper is organized as follows: In Section 2, we briefly review related work. Section 3 introduces our approach and provides the related training and prediction algorithms. In Section 4, we perform a thorough experimental evaluation of StochLWTA-ML, and compare our findings to the current state-of-the-art. In the final Section 5, we end up with the conclusions of our work, and suggest lines of further research.

## 2 RELATED WORK

### 2.1 LWTA LAYERS IN DEEP LEARNING

LWTA layers are not new in the field of deep learning; see, e.g., Srivastava et al. (2013). Although not much work has been pursued along these lines, the recent works of Panousis et al. (2019; 2021) and Voskou et al. (2021) have spurred some fresh interest in the field. These works have presented alternative implementations of the basic concepts of LWTA units in the context of diverse deep network architectures. Specifically, Panousis et al. (2019) propose a stochastic LWTA formulation which is founded upon the Indian Buffet process (IBP) prior, borrowed from nonparametric statistics; they use this architecture to effect data-driven network compression. In their follow-up paper (Panousis et al., 2021), they exploit the same technique to train adversarially-robust deep networks. On the other hand, Voskou et al. (2021) consider a different incarnation of stochastic LWTA architectures, which relies on sampling the winner from a Categorical posterior, driven from the layer input. This layer architecture is used to replace dense ReLU layers in Transformer networks; it is then shown to yield important benefits in a Sign-Language Translation benchmark.

This paper is different from the previous works in various ways: (i) stochasticity does not stem from utilization of the IBP; we rather adopt an approach similar to Voskou et al. (2021); (ii) we do not use the proposed architecture as a replacement for a specific type of layer in a greater network architecture (Transformer) that remains largely unchanged; instead, we build completely new networks using these layers; (iii) we perform a full Bayesian treatment, by treating network weights as random variables; we do not employ this construction as a means of compressing the weights at prediction time, contrary to Panousis et al. (2019), Voskou et al. (2021); instead, we perform weight sampling at prediction time as a means of improving accuracy; and (iv) for the first time, we examine how these principles perform in the context of deep network-driven ML. Note that, apart from LWTA architectures, other data-driven sparsity models have also been proposed recently, e.g. Lee

et al. (2018), Kessler et al. (2021). However, none of these have been developed or evaluated in the context of an ML setting.

## 2.2 Model-Agnostic Meta-Learning

ML (Schmidhuber, 1987), also referred to as *learning to learn* (Thrun & Pratt, 1998), has the goal of obtaining machine learning models that can learn new tasks with limited availability of data, and via only a few gradient steps, by capitalizing upon information stemming from previously learned tasks. In this context, Finn et al. (2017) suggested a *model-agnostic* algorithm for ML, that can be applied to any model trained via gradient descent. The introduced MAML algorithm initializes model parameters in a way that can be quickly adapted to several types of new tasks.

Let us consider a model with parameters $\boldsymbol{\theta}$ and a parametric form $f_{\boldsymbol{\theta}}$. When the model is adapting to an unseen task $T_i$, sampled from the distribution over tasks $P(T)$, MAML initially runs few steps of *inner-loop* gradient descent that yields the task-specific parameter set

$$\boldsymbol{\theta}'_i = \boldsymbol{\theta} - \alpha \nabla_{\boldsymbol{\theta}} L_{T_i}(f_{\boldsymbol{\theta}}) \tag{1}$$

where $\alpha$ is the step size hyperparameter and $L_{T_i}$ denotes the loss on the task $T_i$.

Subsequently, training proceeds to optimize the function $f_{\boldsymbol{\theta}'}$ with respect to the model parameters $\boldsymbol{\theta}$. Assuming that the batch of tasks has size $M$, we can define the targeted *meta-objective* as:

$$L_{meta}(\boldsymbol{\theta}) = \sum_{i=1}^{M} L_{T_i}(f_{\boldsymbol{\theta}'_i}) = \sum_{i=1}^{M} L_{T_i}(f_{\boldsymbol{\theta} - \alpha \nabla_{\boldsymbol{\theta}} L_{T_i}(f_{\boldsymbol{\theta}})}). \tag{2}$$

Optimization of this meta-objective over $\boldsymbol{\theta}$ yields the *outer-loop* update:

$$\boldsymbol{\theta} \leftarrow \boldsymbol{\theta} - \beta \nabla_{\boldsymbol{\theta}} \sum_{i=1}^{M} L_{T_i}(f_{\boldsymbol{\theta}'_i}) \tag{3}$$

where $\beta$ stands for the *outer-loop* learning rate.

Finally, as *MAML* involves expensive computations stemming from the second-order updates of Eqs. (2) and (3), Finn et al. (2017) developed a first-order approximation that reduces the *outer-loop* update (3) to:

$$\boldsymbol{\theta} \leftarrow \boldsymbol{\theta} - \beta \sum_{i=1}^{M} \nabla_{\boldsymbol{\theta}'_i} L_{T_i}(f_{\boldsymbol{\theta}'_i}) \tag{4}$$

In other words, *FOMAML* computes the gradients with respect to the updated parameter values $\boldsymbol{\theta}'_i$, but omits the gradients of $\boldsymbol{\theta}'_i$ with respect to $\boldsymbol{\theta}$.

In a different fashion, the *Reptile* algorithm of Nichol et al. (2018) disposes the costly second-order updates of MAML by: (i) applying some *inner-loop* gradient descent steps using (1); and (ii) suggesting a new *outer-loop* that simply subtracts the parameters, $\boldsymbol{\theta}$, from the updates $\boldsymbol{\theta}'_i$ (instead of computing derivatives):

$$\boldsymbol{\theta} \leftarrow \boldsymbol{\theta} + \beta(\boldsymbol{\theta}'_i - \boldsymbol{\theta}). \tag{5}$$

## 3 Proposed Approach

### 3.1 Architecture

Let us denote as $\boldsymbol{x} \in \mathbb{R}^I$ an input vector presented to a dense ReLU layer of a conventional deep neural network, with corresponding weights matrix $\boldsymbol{W} \in \mathbb{R}^{I \times O}$. The output of the layer is the vector $\boldsymbol{y} \in \mathbb{R}^O$ and is fed to the subsequent layer. In our approach, a ReLU unit is replaced by $J$ competing linear units, organized in one block; in the following, we denote with $R$ the number of blocks in a layer. The input $\boldsymbol{x}$ is now presented to each block through weights that are organized into a three-dimensional matrix $\boldsymbol{W} \in \mathbb{R}^{I \times R \times J}$. Then, the $j$-th competing unit within $r$-th block computes the sum $\sum_{i=1}^{I} (w_{i,r,j}) \cdot x_i$. Competition means that, of the $J$ units in the block, one unit (the "winner") will present its linear computation to the next layer; the rest will present zero

values. Traditionally in the literature, the winner unit is selected to be the unit with greatest linear computation. Recently, stochastic competition principles have been considered, e.g. Panousis et al. (2019; 2021), Voskou et al. (2021).

Let us denote as $\boldsymbol{y} \in \mathbb{R}^{R \cdot J}$ the output of an LWTA layer; this is composed of $R$ subvectors $\boldsymbol{y}_r \in \mathbb{R}^J$ and is sparse, since all units except for one, in each block, yield zero values. Let us introduce the discrete latent indicator vector $\boldsymbol{\xi} \in \text{one\_hot}(J)^R$ to denote the winner units in the $R$ blocks that constitute a considered stochastic LWTA layer. This vector comprises $R$ component subvectors, where each component entails one non-zero value at the index position that corresponds to the winner unit of the respective LWTA block. On this basis, the output $\boldsymbol{y}$ of the stochastic LWTA layer's $(r \cdot j)$-th component $\boldsymbol{y}_{r,j}$ is defined as:

$$\boldsymbol{y}_{r,j} = \boldsymbol{\xi}_{r,j} \sum_{i=1}^{I} (w_{i,r,j}) \cdot x_i \in \mathbb{R} \tag{6}$$

where we denote as $\boldsymbol{\xi}_{r,j}$ the $j$-th component of $\boldsymbol{\xi}_r$, and $\boldsymbol{\xi}_r \in \text{one\_hot}(J)$ holds the $r$-th subvector of $\boldsymbol{\xi}$.

In Eq. (6), we postulate that the latent winner indicator variables are drawn from a Categorical distribution which is proportional to the intermediate linear computation that each unit performs. Therefore, the stronger the linearity the higher the chance of the unit winning the stochastic competition within its block. In detail, we postulate that, a posteriori, the winner distributions yield:

$$q(\boldsymbol{\xi}_r) = \text{Categorical} \left( \boldsymbol{\xi}_r \Big| \text{softmax}(\sum_{i=1}^{I} [w_{i,r,j}]_{j=1}^{J} \cdot x_i) \right) \tag{7}$$

where $[w_{i,r,j}]_{j=1}^{J}$ denotes the vector concatenation of the set $\{w_{i,r,j}\}_{j=1}^{J}$. A graphical illustration of the proposed stochastic architecture is provided in Fig. 1.

As a network composed of such (StochLWTA) layers entails latent variables $\boldsymbol{\xi}$, we need to perform a Bayesian network treatment to perform effective parameter training. We opt for a (approximate) stochastic gradient variational Bayes treatment (Kingma & Welling, 2014), for scalability purposes. This means that the used objective function takes the form of an evidence lower-bound (ELBO) objective, as we describe next. In our work, we take one step further: we also elect to infer a posterior density over the network weights $\boldsymbol{W}$, as opposed to obtaining point-estimates. This results in a second source of stochasticity for our approach, which may further increase its generalization capacity under uncertain conditions arising from limited training data availability. Note that the use of these posteriors is totally different from Panousis et al. (2019) and Voskou et al. (2021): therein, posterior variance is used for compressing posterior mean bit-precision; then, predictions are performed using only the compressed posterior mean. Instead, in our work we sample multiple times from the trained weight posterior and perform model averaging (in a Bayesian sense), as a means of increasing generalization capacity.

We postulate:
$$q(\text{vec}(\boldsymbol{W})) = N(\text{vec}(\boldsymbol{W})|\boldsymbol{\mu}, \text{diag}(\boldsymbol{\sigma}^2)) \tag{8}$$

where $\psi \triangleq \{\boldsymbol{\mu}, \boldsymbol{\sigma}^2\}$ are the means and variances of the Gaussian weight posteriors, respectively.

### 3.2 A Model-Agnostic ML Algorithm

Let us define an ML problem where we are given a training dataset $D$ of tasks, $T$, that are governed by a distribution $P(T)$. We consider an $N$-way, $K$-shot learning setting, where each task contains $K$ labelled examples for each of $N$ available classes.

Our approach entails parameter sets $\boldsymbol{\theta}$ which coincide with the hyperparameter sets $\psi = \{\boldsymbol{\mu}, \boldsymbol{\sigma}^2\}$ of the weights $\boldsymbol{W} \in \mathbb{R}^{I \times R \times J}$; these sets $\psi = \{\boldsymbol{\mu}, \boldsymbol{\sigma}^2\}$ are the target of our MAML-type training algorithm.

Therefore, to perform model training, we have to first initialize the trainable parameters $\boldsymbol{\mu}$ and $\boldsymbol{\sigma}^2$ across layers. To this end, one can appropriately exploit popular initialization schemes, such as

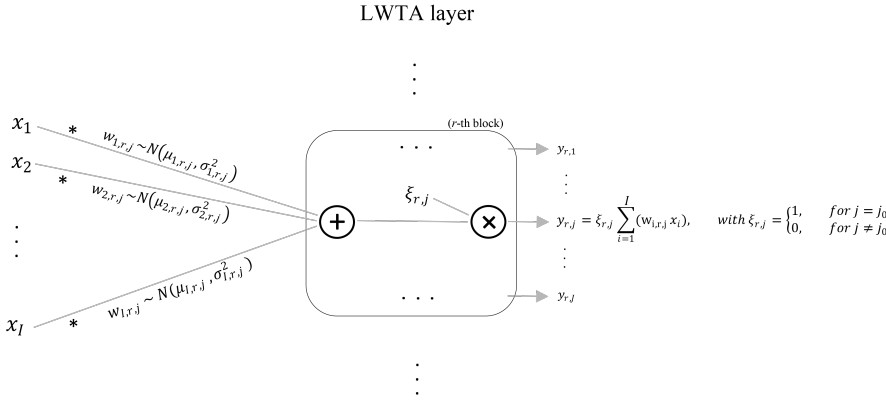

Figure 1: A zoomed-in graphical illustration of the $r$-th block of a stochastic LWTA layer. Input $\boldsymbol{x} = [x_1, x_2, \ldots, x_I]$ is presented to each unit in the block. Assume that the index of the winner unit is $j_0$. Then, the output of the block is a vector with a single non-zero value at index $j_0$.

Glorot Uniform (Glorot & Bengio, 2010). Then, our approach entails an inner-outer loop scheme, in a vein similar to existing approaches, but with some crucial differences:

1. The stochastic nature of the postulated weights, $\boldsymbol{W}$, results in the updates taking place over the posterior means, $\boldsymbol{\mu}$ and variances, $\boldsymbol{\sigma}^2$.

2. The stochastic nature of both the inferred representations and the network weights themselves implies that proper training must rely on optimization of the ELBO function of the network. Let us consider an inner-loop dealing with task $T_i \sim P(T)$, with data $D_i = (X_i, Y_i) \subset D$. Let $\mathrm{CE}(Y_i, f_{\boldsymbol{\psi}}(X_i; \hat{\boldsymbol{\xi}}, \hat{\boldsymbol{W}}))$ be the categorical cross-entropy between the data labels $Y_i$ and the class probabilities $f_{\boldsymbol{\psi}}(X_i; \hat{\boldsymbol{\xi}}, \hat{\boldsymbol{W}})$ generated by the penultimate Softmax layer. Then, we yield:

$$L_{T_i}(\boldsymbol{\psi}) = -\mathrm{CE}(Y_i, f_{\boldsymbol{\psi}}(X_i; \hat{\boldsymbol{\xi}}, \hat{\boldsymbol{W}})) - KL[\,q(\boldsymbol{\xi})\,||\,p(\boldsymbol{\xi})\,] - KL[\,q(\boldsymbol{W})\,||\,p(\boldsymbol{W})\,] \quad (9)$$

for task $T_i$ with dataset $D_i$ which is dealt with on the $i$-th training iteration.

Here, for simplicity and without harming generality, we consider that the weights prior $p(\mathrm{vec}(\boldsymbol{W}))$ is a Gaussian distribution $N(\boldsymbol{0}, \boldsymbol{I})$ and the latent variables prior $p(\boldsymbol{\xi})$ is a symmetric Categorical distribution $\mathrm{Categorical}(1/J)$. In addition, in our notation we stress that the output $f_{\boldsymbol{\psi}}(X_i; \hat{\boldsymbol{\xi}}, \hat{\boldsymbol{W}})$ depends on the winner selection process, which is stochastic, and the outcomes of sampling the network weights. Specifically, in our work we perform Monte-Carlo (MC) sampling using one reparameterized sample of the corresponding latent variables.

Let $\bar{\boldsymbol{\xi}}$ be the unnormalized probabilities of the Categorical distribution $q(\boldsymbol{\xi})$ (Eq. (7)). The sampled instances of $\boldsymbol{\xi}$, $\hat{\boldsymbol{\xi}}_{r,j}$, are expressed as (Maddison et al., 2017):

$$\hat{\boldsymbol{\xi}}_{r,j} = \mathrm{Softmax}((\log \bar{\boldsymbol{\xi}}_{r,j} + g_{r,j})/\tau), \ \forall r = 1, \ldots, R, \ j = 1, \ldots, J \quad (10)$$

where $g_{r,j} = -\log(-\log U_{r,j})$, $U_{r,j} \sim \mathrm{Uniform}(0, 1)$, and $\tau \in (0, \infty)$ is a temperature factor that controls how closely the Categorical distribution is approximated by this continuous relaxation.

Similarly, the Gaussian weights yield: $\hat{w}_{t,r,j} = \mu_{t,r,j} + \sigma_{t,r,j}\hat{\epsilon}$, and $\hat{\epsilon} \sim N(0, 1)$.

On this basis, the KL divergences in Eq. (9) become:

$$KL[\,q(\boldsymbol{\xi}_{r,j}\,||\,p(\boldsymbol{\xi}_{r,j})\,] = \mathbb{E}_{q(\boldsymbol{\xi}_{r,j})}[\log q(\boldsymbol{\xi}_{r,j}) - \log p(\boldsymbol{\xi}_{r,j})]$$
$$\approx \log q(\hat{\boldsymbol{\xi}}_{r,j}) - \log p(\hat{\boldsymbol{\xi}}_{r,j}), \ \forall r, \, j \quad (11)$$

and

$$KL[\,q(w_{t,r,j})\,||\,p(w_{t,r,j})\,] = \mathbb{E}_{q(w_{t,r,j})}[\log q(w_{t,r,j}) - \log p(w_{t,r,j})]$$
$$\approx \log q(\hat{w}_{t,r,j}) - \log p(\hat{w}_{t,r,j}), \ \forall t = 1, \ldots, I, \, r, \, j \quad (12)$$

Hence, the ELBO becomes:

$$L_{T_i}(\boldsymbol{\psi}) = -\mathrm{CE}(Y_i, f_{\boldsymbol{\psi}}(X_i; \hat{\boldsymbol{\xi}}, \hat{\boldsymbol{W}})) - \sum_{r,j} \left( \log q(\hat{\boldsymbol{\xi}}_{r,j}) - \log p(\hat{\boldsymbol{\xi}}_{r,j}) \right) - \sum_{t,r,j} \left( \log q(\hat{w}_{t,r,j}) - \log p(\hat{w}_{t,r,j}) \right)$$

(13)

Therefore, we establish a MAML-type algorithm, where: (i) the used networks comprise blocks of stochastic LWTA units visually depicted in Fig. 1; (ii) the trainable parameters are the means $\boldsymbol{\mu}$ and variances $\boldsymbol{\sigma}^2$ of the synaptic weights; and (iii) the objective function of the inner-loop process is given in Eq. (13).

We summarize our training algorithm in Alg. 1.

---

**Algorithm 1:** Model training with StochLWTA-ML

---

**Require:** $P(T)$: distribution over tasks
Initialize $\boldsymbol{\psi} := \{\boldsymbol{\mu}, \boldsymbol{\sigma}^2\}$
Define outer-step size $\beta$ and inner learning rate $\alpha$
**for** $i = 1, 2, \dots$ **do**
    **Inner training loop**:
        Sample task $T_i \sim P(T)$, where $T_i$ contains data $D_i = (X_i, Y_i) \subset D$
        Compute $L_{T_i}(\boldsymbol{\psi})$ using Eq. (13)
        Compute adapted parameters with SGD: $\boldsymbol{\psi}'_i = \boldsymbol{\psi} - \alpha \nabla_{\boldsymbol{\psi}} L_{T_i}(f_{\boldsymbol{\psi}})$
    **Outer training loop**:
        Derive $\boldsymbol{\psi} \leftarrow \boldsymbol{\psi} + \beta(\boldsymbol{\psi}'_i - \boldsymbol{\psi})$
**end**

---

### 3.3 PREDICTION ALGORITHM

At prediction time, we draw a set of $B$ samples of the Gaussian connection weights from the trained posteriors $\mathcal{N}(\boldsymbol{\mu}, \boldsymbol{\sigma}^2)$. Then, we select the winning units in each block of the network by similarly sampling from the posteriors $q(\boldsymbol{\xi})$. This results in a set of $B$ output logits of the network, which we average to obtain the final predictive outcome:

$$f_{\boldsymbol{\psi}}(X_i; \tilde{\boldsymbol{\xi}}, \tilde{\boldsymbol{W}}) \approx \frac{1}{B} \sum_{s=1}^{B} f_{\boldsymbol{\psi}}(X_i; \tilde{\boldsymbol{\xi}}_s, \tilde{\boldsymbol{W}}_s)$$

(14)

where $\tilde{\boldsymbol{\xi}}_s$ and $\tilde{\boldsymbol{W}}_s$ are sampled directly from the posteriors $q(\boldsymbol{\xi})$ and $q(\boldsymbol{W})$, respectively.

This concludes the formulation of the proposed model-agnostic ML approach.

---

**Algorithm 2:** Prediction with StochLWTA-ML

---

**Require:** Learned parameters $\boldsymbol{\psi} = \{\boldsymbol{\mu}, \boldsymbol{\sigma}^2\}$, input data $X'$
    Sample $\boldsymbol{W} \sim q(\mathrm{vec}(\boldsymbol{W})) = N(\mathrm{vec}(\boldsymbol{W})|\boldsymbol{\mu}, \mathrm{diag}(\boldsymbol{\sigma}^2))$
    Sample $\boldsymbol{\xi} \sim q(\boldsymbol{\xi})$ defined in Eq. (7), for $\boldsymbol{w} = \boldsymbol{\psi}$ and $(x_i = x'_i) \in X'$
    Compute output logits, given the sampled values $\boldsymbol{\xi}$ and $\boldsymbol{\psi}$
**Repeat** $B$ times
Use Eq. (14) to average over the resulting set of $B$ logits and derive the final prediction.

---

## 4 EXPERIMENTS

### 4.1 EXPERIMENTAL SETUP

We evaluate StochLWTA-ML on Omniglot, Mini-Imagenet and CIFAR-100 which are popular few-shot image classification datasets, and compare its performance to state-of-the-art prior results. After thorough exploration on the number of LWTA layers as well as the number of blocks for each layer and the competing units per block, we end up with using networks comprising 2 layers with 16 blocks and 2 competing units per block on the former layer, and 8 blocks with 2 units per block on the latter. The penultimate network layer is a Softmax. Weight mean initialization, as well as point-estimate initialization for our competitors, is performed via Glorot Uniform. Weight log-variance initialization is performed via Glorot Normal, by sampling from $N(0.0005, 0.01)$. The Gumbel-Softmax relaxation temperature is set to $\tau = 0.67$.

In the inner-loop updates, we use the Stochastic Gradient Descent (SGD) (Robbins, 2007) optimizer with a learning rate of 0.003. For the outer-loop, we use SGD with a linear annealed outer step size to 0, and an initial value of 0.25. Additionally, all the experiments were ran with task batch size of 50 for both training and testing mode. Prediction is carried out averaging over $B = 4$ output logits.

The results presented in Sections 4.2 and 4.3 stand for the average performance over three runs with different random seeds. Note that each experiment consists of different number of iterations for training, depending on its convergence speed. The code was implemented in Tensorflow (Abadi et al., 2016).

## 4.2 RESULTS

In Table 1, we show how StochLWTA-ML performs on Omniglot 20-way, Mini-Imagenet 5-way and CIFAR-100 5-way few-shot settings. We compare our findings to state-of-the-art ML algorithms such as LLAMA (Grant et al., 2018) and PLATIPUS (Finn et al., 2018) as reported in Gordon et al. (2018), Amortized Bayesian Meta-Learning (ABML) (Ravi & Beatson, 2019), MAML, FOMAML (Finn et al., 2017), Reptile (Nichol et al., 2018) and others. Using the original architectures with the same hyperparameters and data preprocessing as in Finn et al. (2017), we have also locally reproduced ABML, BMAML (with 5 particles), PLATIPUS, MAML, FOMAML and Reptile (dubbed "local" in Table 1). For completeness sake, we also compare our findings to other state-of-the-art ML models as reported in Finn et al. (2017), including Matching Nets (Santoro et al., 2016) and LSTM Meta-Learner (Ravi & Larochelle, 2017). As we observe, our method outperforms the existing state-of-the-art in both the 1-shot and 5-shot settings.

Table 1: N-way K-shot (%) classification accuracies on Omniglot, Mini-Imagenet and CIFAR-100

| | Omniglot 20-way | | Mini-Imagenet 5-way | | CIFAR-100 5-way | |
|---|---|---|---|---|---|---|
| **Algorithm** | 1-shot | 5-shot | 1-shot | 5-shot | 1-shot | 5-shot |
| Matching Nets | 93.80 | 98.50 | 43.56 | 55.31 | - | - |
| LSTM Meta-Learner | - | - | 43.44 | 60.60 | - | - |
| MAML | 95.80 | 98.90 | 48.70 | 63.11 | - | - |
| FOMAML | - | - | 48.07 | 63.15 | - | - |
| Reptile | 88.14 | 96.65 | 47.07 | 62.74 | - | - |
| PredCP (Nalisnick et al., 2021) | - | - | 49.30 | 61.90 | - | - |
| Neural Statistician (Edwards & Storkey, 2016) | 93.20 | 98.10 | - | - | - | - |
| mAP-SSVM (Triantafillou et al., 2017) | 95.20 | 98.60 | 50.32 | 63.94 | - | - |
| LLAMA | - | - | 49.40 | - | - | - |
| PLATIPUS | - | - | 50.13 | - | - | - |
| GEM-BML+ (Zou & Lu, 2020) | 96.24 | 98.94 | 50.03 | - | - | - |
| DKT (Patacchiola et al., 2020) | - | - | 49.73 | 64.00 | - | - |
| ABML | - | - | 45.00 | - | 49.50 | - |
| BMAML (with 5 particles) (Yoon et al., 2018) | - | - | 53.80 | - | - | - |
| ABML (local) | 90.21 | 93.39 | 44.23 | 52.12 | 49.23 | 53.60 |
| BMAML (local) | 96.92 | 98.11 | 53.10 | 64.80 | 52.60 | 65.80 |
| PLATIPUS (local) | 94.35 | 98.30 | 49.97 | 63.13 | 51.14 | 63.61 |
| MAML (local) | 95.48 | 98.61 | 48.60 | 63.01 | 50.67 | 62.89 |
| FOMAML (local) | 94.92 | 98.12 | 47.93 | 63.10 | 49.13 | 63.80 |
| Reptile (local) | 87.98 | 96.36 | 46.97 | 62.53 | 48.19 | 63.45 |
| **StochLWTA-ML** | **97.79** | **98.97** | **54.11** | **66.70** | **54.60** | **66.73** |

## 4.3 ABLATION STUDY

### 4.3.1 DOES STOCHASTIC COMPETITION CONTRIBUTE TO CLASSIFICATION ACCURACY?

To check whether the accuracy improvements stem from the LWTA-induced sparsity or the proposed stochastic competition concept, we evaluate both our approach as well as MAML, FOMAML, ABML, BMAML and PLATIPUS, considering both "deterministic LWTA" and "stochastic LWTA" setups; deterministic LWTA networks have been adopted from Srivastava et al. (2013). As we see in Table 2, replacing ReLU with deterministic LWTA yields negligible differences. On the other hand, stochastic LWTA units yield a clear improvement in all cases. This improvement becomes even more important in the case of our approach, where we sample from stochastic weights.

Table 2: Mini-Imagenet 5-way Few-Shot ablation study (% accuracy)

| Algorithm | Network type | 1-shot | 5-shot |
|---|---|---|---|
| MAML (local) | deterministic LWTA | 48.88 | 63.15 |
| | stochastic LWTA | 49.61 | 64.03 |
| FOMAML (local) | deterministic LWTA | 48.11 | 63.54 |
| | stochastic LWTA | 49.24 | 64.54 |
| ABML (local) | deterministic LWTA | 44.31 | 52.27 |
| | stochastic LWTA | 45.11 | 53.31 |
| BMAML (local) | deterministic LWTA | 53.12 | 64.84 |
| | stochastic LWTA | 53.50 | 65.31 |
| PLATIPUS (local) | deterministic LWTA | 49.99 | 63.21 |
| | stochastic LWTA | 51.06 | 64.18 |
| **StochLWTA-ML** | deterministic LWTA | 53.12 | 64.93 |
| | stochastic LWTA | **54.11** | **66.70** |

### 4.3.2 EFFECT OF BLOCK SIZE $J$

As it is presented in Table 3, increasing the number of competing units per block to $J = 4$ or $J = 8$ does not notably improve the results of our approach. On the contrary, it increases the number of trained parameters, thus leading to higher network computational complexity. This corroborates our initial choice of using $J = 2$ competing units per block in our approach.

Table 3: Effect of block size $J$ in StochLWTA-ML's classification (%) accuracy

| | Omniglot 20-way | | Mini-Imagenet 5-way | | CIFAR-100 5-way | |
|---|---|---|---|---|---|---|
| **Number of units** | 1-shot | 5-shot | 1-shot | 5-shot | 1-shot | 5-shot |
| $J = 2$ | 97.79 | 98.97 | 54.11 | 66.70 | 54.60 | 66.73 |
| $J = 4$ | 96.33 | 98.55 | 53.99 | 66.65 | 54.51 | 66.13 |
| $J = 8$ | 95.38 | 98.83 | 53.70 | 67.08 | 54.45 | 66.18 |

### 4.3.3 HOW DOES THE NUMBER OF SAMPLES AT PREDICTION TIME AFFECT ACCURACY?

We scrutinize the effect of the number of drawn samples, $B$, on StochLWTA-ML's predictive accuracy. To this end, we repeat our experiments using $B = 10$ logits sets. In Table 4, we provide the comparative outcomes concerning the two sample size configurations of $B = 4$ and $B = 10$. As we observe, an increase in prediction sample size, $B$, yields a slight accuracy increase. However, the aforementioned increase in sample size imposes some computational overhead, which we elaborate upon in the following Section. We argue that this overhead might not be worth it for the slight performance increase reported in Table 4. More information on the effect of sample size in our approach's predictive performance are provided in the Supplementary.

Table 4: Effect of sample size $B$ in StochLWTA-ML's classification (%) accuracy

| | Omniglot 20-way | | Mini-Imagenet 5-way | | CIFAR-100 5-way | |
|---|---|---|---|---|---|---|
| **Number of samples** | 1-shot | 5-shot | 1-shot | 5-shot | 1-shot | 5-shot |
| $B = 4$ | 97.79 | 98.97 | 54.11 | 66.70 | 54.60 | 66.73 |
| $B = 10$ | 96.91 | 99.23 | 54.89 | 67.22 | 55.18 | 66.15 |

### 4.3.4 IS THERE A COMPUTATIONAL TIME TRADE-OFF FOR THE INCREASED ACCURACY?

It is also important to investigate whether our approach represents a trade-off between accuracy and computational time compared to our competitors. To facilitate this investigation, in Table 5 we provide training iteration wall-clock times for our approach and the existing locally reproduced state-of-the-art, as well as the total number of iterations each model needs to achieve the reported performance of Table 1. It appears that our methodology takes 77% *less* training time than the less efficient algorithms ABML, BMAML, PLATIPUS, and is comparable to other approaches. This happens because our approach yields the reported state-of-the-art performance by employing a network architecture (that is, number of LWTA layers, as well as number of blocks and block size on each layer) that result in a total number of trainable parameters that is *one order of magnitude less* on average than the best performing baseline methods. This can be seen in the last three columns of Table

[5](#) (dubbed $D_A$, $D_B$ and $D_C$ for Omniglot, Mini-Imagenet and CIFAR-100 respectively). In addition, training for our approach converges fast.

The situation changes when it comes to prediction: our approach imposes a slight computational time overhead compared to MAML, FOMAML and Reptile, but still *much less* than the time-consuming PLATIPUS. BMAML and ABML. This is a rather negligible increase when we are dealing with a low number of drawn samples, $B$, but increases as we increase $B$. The provided results suggest that a selection of $B = 4$ represents a favorable accuracy/prediction wall-clock time for our approach.

Table 5: Performance comparison: average wall-clock time (in msecs), training iterations for each locally reproduced method and number of baselines' trainable parameters over the considered datasets of Table [1](#)

| Algorithm | Training | Prediction | Number of training iterations | $D_A$ parameters | $D_B$ parameters | $D_C$ parameters |
|---|---|---|---|---|---|---|
| PLATIPUS (local) | 1603.39 | 602.77 | 333600 | 560025 | 615395 | 580440 |
| BMAML (local) | 1450.31 | 514.43 | 301800 | 560025 | 615395 | 580440 |
| ABML (local) | 678.48 | 265.78 | 138000 | 224010 | 246158 | 232176 |
| MAML (local) | 288.25 | 103.28 | 60000 | 112005 | 123079 | 116088 |
| FOMAML (local) | 284.49 | 102.34 | 60000 | 112005 | 123079 | 116088 |
| Reptile (local) | 284.30 | **102.27** | 60000 | 113221 | 124613 | 117463 |
| **StochLWTA-ML** | **282.90** | 113.44 ($B = 4$) 121.87 ($B = 10$) | 60000 | **54549** | **60112** | **56745** |

Finally, we provide an example of how training for our approach converges, and how this compares to the alternatives. We illustrate our outcomes on the Omniglot 20-way 1-shot benchmark; similar outcomes have been observed in the rest of the considered datasets. Fig. [2](#)(a) compares StochLWTA-ML with prior traditional ML methods: MAML, FOMAML and Reptile. It becomes apparent that our approach converges equally fast to these competitors. Further, Fig. [2](#)(b) compares StochLWTA-ML with the probabilistic ML models ABML, BMAML, PLATIPUS. Since, as we see in the ablation study of Section [4.3.4](#), these methods are quite time-consuming and less efficient regarding to memory consumption, StochLWTA-ML gives rise to an easier time training MAML based probabilistic model.

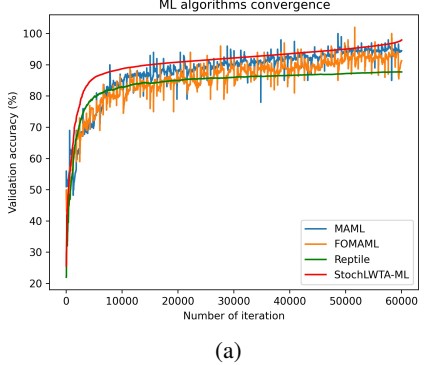
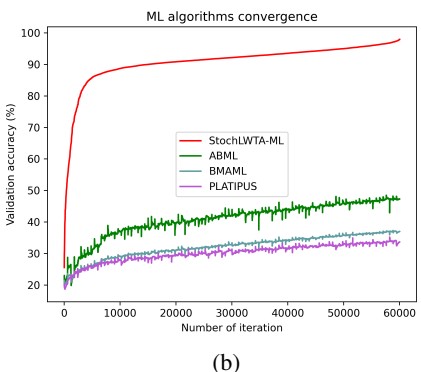

(a)                                                    (b)

Figure 2: ML algorithms' training convergence comparison

# 5 CONCLUSION

In this paper, we proposed a sparse and stochastic network paradigm for ML, with novel network design principles compared to currently used model-agnostic ML models. We introduced *stochastic LWTA activations* in the context of a *variational Bayesian treatment* that gave rise to a doubly-stochastic ML framework, bearing the promise of stronger generalization capacity. We evaluated our approach using standard benchmarks in the field, and showed that it outperformed the state-of-the-art in terms of both predictive accuracy and computational costs. The results have provided strong empirical evidence supporting our claims. In the future, we plan to study the effect of StochLWTA-ML in areas related to ML, such as Continual Learning (Javed & White, 2019) and Reinforcement Learning (Zhu et al., 2020).

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

## A    FURTHER DETAILS ON DATASETS EXPERIMENTAL FORMATION

Omniglot is a dataset of 1623 characters from different alphabets, containing 20 examples per character scaled down to 28x28 grayscale pixels. The ratio between training and testings sets is 3:2, so after shuffling the character classes we randomly choose the first 974 classes for training and the remaining are left for testing. As for the Mini-Imagenet dataset, it has color images of size 84x84 and contains 100 classes with 600 examples from the ImageNet dataset. We randomly choose 45000 examples for the training phase and the rest constitute the testing population. The CIFAR-100 dataset consists of color images of size 32x32 and contains 100 classes with 600 images per class. We randomly choose 500 images per class for training and the rest 100 images per class constitute the testing population.

## B    ADDITIONAL EXPERIMENTS

In Table B1 we provide the results of the ablation study of Section 4.3.1 on the Omniglot 20-way dataset. The full-fledged StochLWTA-ML approach yields again better predictive performance compared to the alternative variants of ML algorithms.

Table B1: Omniglot 20-way Few-Shot ablation study (% accuracy)

| Algorithm | Network type | 1-shot | 5-shot |
|---|---|---|---|
| MAML (local) | deterministic LWTA | 95.52 | 98.15 |
| | stochastic LWTA | 95.91 | 98.78 |
| FOMAML (local) | deterministic LWTA | 95.01 | 98.18 |
| | stochastic LWTA | 95.80 | 98.41 |
| ABML (local) | deterministic LWTA | 90.30 | 93.64 |
| | stochastic LWTA | 91.21 | 93.91 |
| BMAML (local) | deterministic LWTA | 96.96 | 98.21 |
| | stochastic LWTA | 97.11 | 98.30 |
| PLATIPUS (local) | deterministic LWTA | 94.48 | 98.31 |
| | stochastic LWTA | 95.13 | 98.56 |
| **StochLWTA-ML** | deterministic LWTA | 96.95 | 98.63 |
| | stochastic LWTA | **97.79** | **98.97** |

## C    FEW-SHOT CLASSIFICATION NETWORK ARCHITECTURES

For the local replicates of prior ML algorithms in the experiments of our work, we follow the same architecture for the deep neural network as the one used by Vinyals et al. (2016). For Omniglot, the network is composed of 4 convolutional layers with 64 filters, 3 x 3 convolutions and 2 x 2 strides, followed by a Batch Normalization layer (Ioffe & Szegedy, 2015) and the final values of each layer are processed by an activation function. For both Mini-Imagenet and CIFAR-100, we use 4 convolutional layers with 32 filters to reduce overfitting like Ravi & Larochelle (2017), 3 x 3 convolutions followed by Batch Normalization layer and $2 \times 2$ max-pooling layer with the values of each layer finally passed again through an activation block. The activation functions used for experiments of main paper's Tables 1 and 5 is ReLU, and LWTA for experiments of main paper's Table 2 and Supplementary's Table B1.

# D  HOW DOES THE TASK BATCH SIZE AFFECT STOCHLWTA-ML'S PERFORMANCE?

In Fig. 3 and 4, we illustrate the performance of our model on the Mini-Imagenet 5-way 1-shot setting with different task batch sizes. As we see, our model performs optimally with task batch size of 50 for both training and testing phase. Using this value for batch size, we noticed that the classification accuracy as well as training time per iteration were optimal. This outcome was also observed in the rest of the considered datasets' settings.

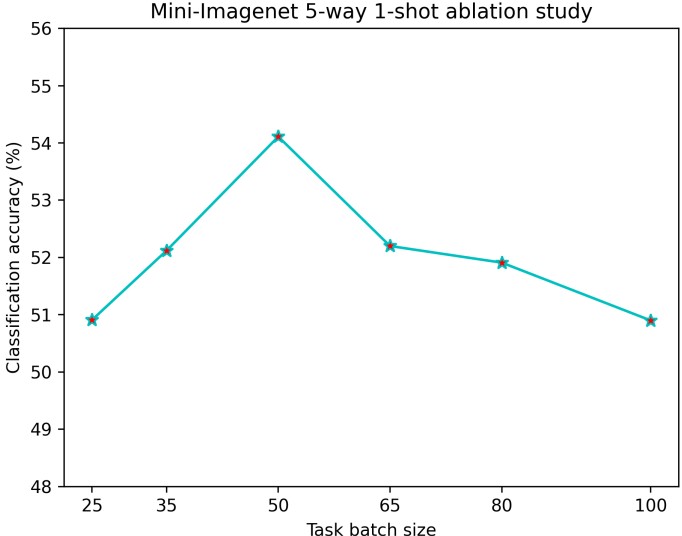

Figure 3: The effect of task batch size in StochLWTA's predictive accuracy

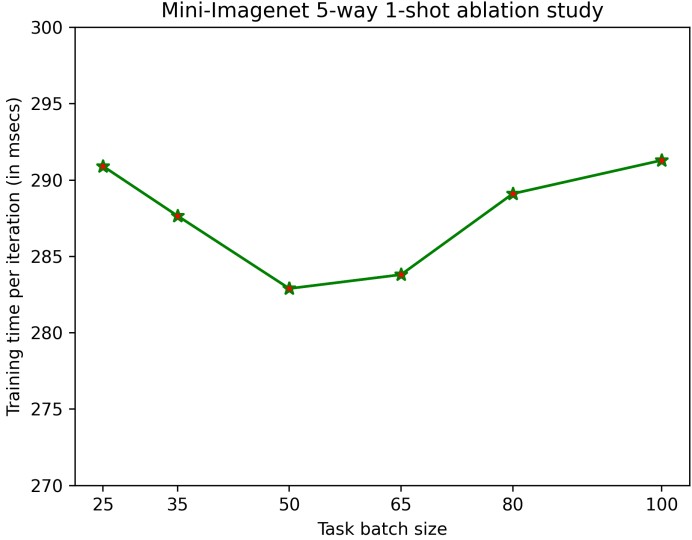

Figure 4: The effect of task batch size in StochLWTA's training time per iteration (in msecs)

# E HOW DOES THE SAMPLE SIZE $B$ AT PREDICTION TIME AFFECT STOCHLWTA-ML'S ACCURACY?

As we observe in Fig. 5, an increase in sample size, $B$, does not always yield an accuracy increase. We finally choose $B = 4$, since at that point our model achieves its best predictive performance at most of the experiments. In this figure, we illustrate our outcomes on Omniglot 20-way 1-shot, Mini-Imagenet 5-way 1-shot and CIFAR-100 5-way 5-shot settings. Our approach has shown similar behaviour to the rest of experimental settings, leading us to the same choice for the number of sample size $B$.

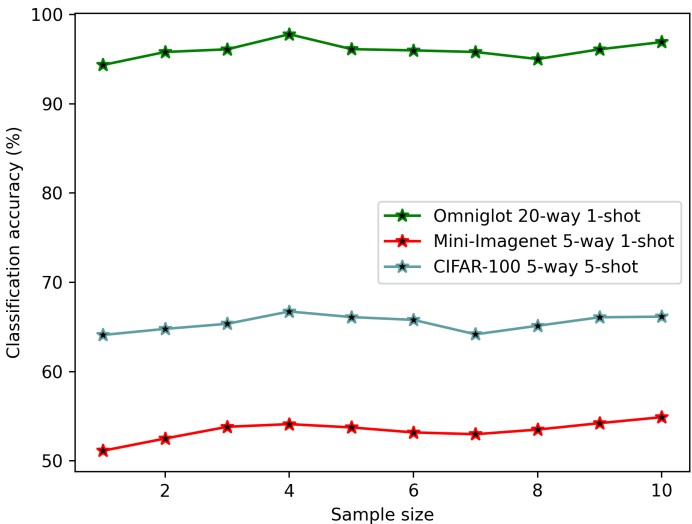

Figure 5: The effect of sample size $B$ in StochLWTA-ML's classification ($\%$) accuracy

# F WHAT PARAMETERS DO WE COUNT FOR THE OUTCOMES OF MAIN PAPER'S TABLE 5?

The included parameters of each baseline for the outcomes of Table 5 are:

- PLATIPUS: $\Theta = \{\boldsymbol{\mu}_\theta, \boldsymbol{\sigma}_\theta^2, \boldsymbol{v}_\theta, \boldsymbol{\gamma}_p, \boldsymbol{\gamma}_q\}$
- BMAML: $\Theta = \{\theta^m\}_{m=1}^5$, for using 5 particles
- ABML: $\theta = \{\boldsymbol{\mu}_\theta, \boldsymbol{\sigma}_\theta^2\}$
- MAML: $\theta = \{\boldsymbol{\mu}_\theta\}$
- FOMAML: $\theta = \{\boldsymbol{\mu}_\theta\}$
- Reptile: $\theta = \{\boldsymbol{\mu}_\theta\}$
- StochLWTA-ML: $\theta = \{\boldsymbol{\mu}_\theta, \boldsymbol{\sigma}_\theta^2\}$

