# OpenReview forum: "Stochastic Deep Networks with Linear Competing Units for Model-Agnostic Meta-Learning"
_ICLR.cc/2022/Conference — ICLR 2022 Submitted_

### Official Review · Reviewer_Nj3T · 2021-10-29

**Correctness:** 2
**Technical Novelty And Significance:** 2
**Empirical Novelty And Significance:** 3
**Recommendation:** 5
**Confidence:** 3

**Main Review:**

1. The novelty of the paper is limited. It applies stochastic LWTA to the standard MAML framework. Using Bayesian Model Averaging for its inferencing is also not new. Also, I do not find any advantage of this stochastic design which in fact induces large computation overhead.

2. This paper lacks insight into how the proposed design is specifically related to meta-learning. Why does the "sparse representation" induced by the LWTA function help?

3. The experiment results are strong. The authors provided a detailed experimental setting for reproducing the result. The improvement of the proposed model over MAML and FOMAML is significant. However, I would still suggest the authors compare their method to other recent state-of-the-art meta-learning methods.

4. The authors provided an ablation study to empirically prove the stochastic LWTA is the underlying force for improving the model. However, I think the ablation study needs to be broken down into a deterministic LWTA function and a stochastic nature. It's not clear actually the model benefits from which part. The authors also discussed the computation overhead. These experiments are limited to small model settings without discussion of the role of batch size in the play.


**Summary Of The Paper:**

This paper proposed a novel meta-learning method by replacing standard nonlinearity functions in MAML with stochastic local winner-takes-all (LWTA) activations. The authors claim that such design results in sparse representations and benefit meta-learning. This method demonstrated superior performance over MAML and FOMAML over standard benchmarks such as Omniglot and Mini-ImageNet.

**Summary Of The Review:**

This paper improves standard MAML by replacing the nonlinearity with stochastic LWTA. The idea has limited novelty and limited insight.
The pros are the experiments are well performed, and the results are significant. I appreciate the ablation study and the overhead complexity discussion.

---

> ### Author Response · Authors · 2021-11-19
> **Response to reviewer Nj3T**
>
> 1) (i) To begin with, see response to reviewer 1, at points 1 and 2. We stress that it is the first time an LWTA activation of stochastic type is examined and $\textit{properly configured}$ in the context of ML. This sort of study transcends simple off-the-shelf application of a technique, as it requires examination of proper configuration, e.g., hyperparameters and priors, treatment of model training, for instance via imposition of appropriate priors and derivation of corresponding posteriors over the weights, etc.
>
>     (ii) In addition, and most crucially, our experimental results show that our stochastic design yields accuracy $\textit{better}$ than the state-of-the-art with trained networks that comprise $\textit{immensely less}$ units, layers, etc, and therefore $\textit{immensely less}$ trainable parameters and associated computational and memory footprint. Please refer to Tables 1 and 5.
>
> 2) Sparsity and stochasticity in deep network representations are generally considered to be auspicious to generalization capacity. In this paper, the goal was to examine whether this can be the case if using stochastic LWTA blocks to obtain these representation properties. The results have been affirmative.
>
> 3) We have added more experiments in Section 4. See point 3 to reviewer 1.
>
> 4)
>     (i) See point (iii) to reviewer 2 in "Somewhat narrow set of main experiments" section.
>
>     (ii) In Section D of the appendix, we have illustrated the performance of our model on the Mini-Imagenet 5-way 1-shot setting with different task batch sizes. As we see, our model performs optimally with task batch size of 50 for both training and testing phase. Using this value for batch size, we noticed that the classification accuracy as well as training time per iteration were optimal. This outcome was also observed in the rest of the considered datasets' settings.

---

### Official Review · Reviewer_rRh2 · 2021-11-02

**Correctness:** 3
**Technical Novelty And Significance:** 3
**Empirical Novelty And Significance:** 3
**Recommendation:** 3
**Confidence:** 3

**Main Review:**

The paper is interesting, but is not very clear. Here are a few examples that I was confused with when I first read the paper. I suggest the authors do a thorough revision to make the paper more approachable.

1. Intuitively, why does LWTA units improve the performance with fewer data?
2. What are the definitions of [\xi]_{r, j} and [\xi]_r? Because of this confusion, I couldn't really understand (9) or (10).
3. What is a "Discrete" distribution? is it a "multinoulli distribution"?
4. Why should the density of [\xi]_r (again, I don't quite understand this notation) be proportional to the size of the activation?
5. It is unclear to me why we need to sample W from a distribution, rather than obtaining its point estimate using ERM?
6. The description of the training procedure on page 6 (especially the paragraph above (13) is very dense. I think I got a big picture of the proposal, but I don't think I can reproduce their methodology based on this description.

**Summary Of The Paper:**

In this paper, the authors proposed to replace the classical activation units in MAML with stochastic local winner-takes-all (LWTA) units. The authors argued the biologically motivated LWTA units would lead to better performance in the few-shot setting where the support set in the target task is small, in which case the embedding representation of the target task would be noisy. The authors framed their proposal in the Bayesian setting, and proposed algorithms to estimate the distribution parameters.

**Summary Of The Review:**

I think the paper is interesting, but the presentation needs to be greatly improved. Many notations are used without introduction, and the description of the methodology is very dense that I am afraid that unless the reader is very familiar with the field, they could not reproduce the methods based on the description

---

> ### Author Response · Authors · 2021-11-19
> **Response to reviewer rRh2**
>
> 1) We posit that these outcomes are due to two main features that differentiate our approach from the baselines we compare against:
>
>     (i) The fact that the obtained representations are sparse, that is they entail many zero values. There is a large corpus of works that have examined the benefits of sparse representations in deep learning.
>
>     (ii) The stochastic nature of the representations, that is the fact that the same input may result in different representations each time it is presented to the network. This property essentially forces the training process to learn to extract representations that are less sensitive to unimportant data properties and generalize better.
>
> (2, 4) Please see responses to reviewer 2, at points 3 and 4. We emphasize that the rationale of setting the probability of winning proportional to the unit's linear computation is a sensible construction that is data-driven, as reviewer 2 expected to see.
>
>
> 3) Yes. "Discrete" distribution is a Categorical/multinoulli distribution.
>
>
> 5) Traditionally, deep networks obtain point-estimates for their parameters. Our model requires stochastic gradient variational Bayes for its training, as it entails the latent variables $\xi$; thus, we have to define an ELBO and reparameterization tricks. Having already made this effort, it made sense to also consider the case of performing a full Bayesian treatment, which essentially means training (variational) posteriors over the weights, instead of point-estimates. The results have been favorable, hence we have retained these in the formulation of our approach.
>
>
> 6) We have rewritten Section 3.2 to make the training process crystal clear.

---

### Official Review · Reviewer_3Mcm · 2021-11-03

**Correctness:** 2
**Technical Novelty And Significance:** 3
**Empirical Novelty And Significance:** 3
**Recommendation:** 5
**Confidence:** 3

**Main Review:**

## Strengths

* Strong empiracle results

* Great set of ablation studies

## Weaknesses

### Lack of novelty

There already exist many probabilistic approaches to meta-learning which take uncertainty into account, which the authors have failed to acknowledge. Thus their claims of novelty in this regard are incorrect. For example, the statement "these innovations constitute a radical departure from the currently prominent design paradigm in ML" is far too strong a statement given the work of Finn et al. (2018), Nalisnick et al. (2021), Chen et al. (2020), and Gordon et al. (2019).

### Somewhat narrow set of main experiments

The main set of results in the paper are based only on the Omniglot 20-way and Mini-imagenet 5-way tasks. While the results on these tasks are good, more tasks are required to paint an accurate picture of the method. Cifar100 few shot would be a good start, but even better would be a non-image classification task. I might suggest the ShapeNet View Reconstruction task as used in Gordon et al. (2019).

Furthermore, at least some of the methods from Finn et al. (2018), Nalisnick et al. (2021), Chen et al. (2020), and Gordon et al. (2019) should be compared against.

An additional ablation study comparing the performance of Stochastic LWTA and standard LWTA would be informative.

For the effect of sample size ablation, plots with performance on the y-axis and B on the x-axis for B in {1,...,10} would be very interesting as they would hopefully show a "knee-point" at which it stops being worthwhile to increase B.

### Lack of clarity

Overall I found the presentation to be confusing in many places and occasionally incorrect. I will list the issues I spotted below.

1. The authors frequently refer to "stochastic arguments", "modelling arguments", "deep network arguments", etc. However, the usage of the word "argument" in these contexts does not make sense to me and I am unsure exactly what is meant.

2. The authors frequently use the word "inference" to refer to test time prediction. However, this word also refers to the probabilistic inference for random variables. Given that the authors' proposed method makes use of such inference this becomes somewhat confusing. I would recommend only using inference in the probabilistic sense.

3. \vec{y}_r is introduced in sec 2.2 but it is not mentioned until much later in sec 3.1 that \vec{y} is the concatenation of \vec{y}_r.

4. It is not clear why it is a good idea to use a fixed competition function that always chooses the maximum activation. It would be great if the authors could provide some motivation/intuition for this choice. (As I mention below, I think that the Sparse-MoE approach with an input-dependent gating network makes more sense).

5. It is not clear what the authors mean by "We posit that this novel stochastic paradigm will give rise to more **potent** learned representations ..." (emphasis my own). (This is another example of a claim that overstates the novelty of this work.)

6. (Minor) I assume that the authors mean "model-agnostic ML" rather than "meta-agnostic ML" at the start of sec 3.1.

7. (Minor) The authors introduce the symbols A_1 and A_2 to indicate the input and output dimensions of their linear units. Would I and O not be more clear?

8. Paragraphs 2 and 3 are largely redundant as they mainly contain repeated information from sec 2.2.

9. (Minor) In eqn 9, the notation [\vec{y}]_{r, j} is used, but just above the notation y = [\vec{y}_{r, j}]. I found this somewhat confusing.

10. In the paragraph below eon 9, the authors first say that they "postulate" that the indicator variable is drawn from a Discrete distribution. Later in the paragraph they say that they "stipulate" this. Which one is it?

11. Speaking of the "Discrete" probability distribution, would it not be more clear to refer to a "Categorical" probability distribution?

12. In the paragraph after eqn 10, the authors state that they impose a posterior density over the weights. However, it is more accurate to say that the prior density is impose and the posterior is inferred.

13. The authors frequently refer to "stochasticity" as increasing the generalisation capacity. However, it should be noted that simply adding stochasticity does not help. It must be added in a principled and theoretically justified manner.

14. "This concludes the formulation of the proposed model-agnostic ML approach." – I suspect this line should be at the end of sec 3.3, rather than sec 3.1.

15. The authors refer to independence among layers in the context of the mean-field approximation. I assume that this was meant to be independence among weights?

16. Eqn 13, for the Gumbel-softmax does not show the purpose of the temperature \tau.

17. In algorithm 2, line 2, the authors suggest that the variational parameters {\mu, \sigma} should be sampled. I suspect that the authors mean \theta or \vec{w}, should be sampled?

17. In algorithm 2, the authors should show the equation for the BMA.

18. At the top of page 7, the authors refer to "2 competing units per block" and then "2 neurons per block". Are neurons the same as competing units?

19. "For the outer-loop, we use SGD with a linear annealed outer step size equal to 0." Should this actually be 0?

20. What is the architecture that was used for the baseline methods?

21. The authors claim a 10% improvement over the "best alternatives" however, I think it is a 10% improvement over the *worst* alternatives.

22. In table 4, it surprises me that Reptile runs slower than MAML. I don't think this makes sense. Also in table 4, the best value in the "inference" column should be bolded.

23. In table 5, does the parameter count for LWTA include both variational parameters \mu and \sigma^2? OR is it just \theta?

24. More motivation is needed for the statement "It appears that convergence of our approach is even smoother; ...".

25. Figure 2b, should be replaced with Figure 2a but for Mini-imagenet.

## Other comments

1. There is a broad range of work involving so-called Sparse Mixture of Experts layers which are conceptually very similar to the stochastic LWTA activated layers. Sparse MoEs have the advantage that they are more computationally efficient. They also have, in my opinion, a more sensible input-dependent gating mechanism compared to simply choosing the most positive activation. These have not been mentioned at all in this work. See (Riquelme et al., 2021) and (Shazeer et al., 2017).

2. I think that the statement "there is an increasing body of evidence from Neuroscience that neurone with similar functions ..." requires a citation. Perhaps (Lasner, 2009) is the source for this statement, but that is not clear.

3. "To ensure statistical significance ..." – providing multiple runs (while greatly appreciated) does not make the results statistically significant. If the authors wish to claim statistical significance, they should perform the appropriate statistical tests.

4. It is not clear to me that the authors' proposed method actually provides better results for the harder Mini-imagenet task. It is true that the raw gain of 10% Mini-imagenet is larger than the 2% for Omniglot. However, because Omniglot accuracy is nearly saturated by the baselines already, this 2% improvement may actually be even more impressive.

## References

Carlos Riquelme, Joan Puigcerver, Basil Mustafa, Maxim Neumann, Rodolphe Jenatton, André Susano Pinto, Daniel Keysers, Neil Houlsby:
Scaling Vision with Sparse Mixture of Experts. CoRR abs/2106.05974 (2021)

Noam Shazeer, Azalia Mirhoseini, Krzysztof Maziarz, Andy Davis, Quoc V. Le, Geoffrey E. Hinton, Jeff Dean:
Outrageously Large Neural Networks: The Sparsely-Gated Mixture-of-Experts Layer. ICLR (Poster) 2017

Chelsea Finn, Kelvin Xu, Sergey Levine:
Probabilistic Model-Agnostic Meta-Learning. NeurIPS 2018: 9537-9548

Eric T. Nalisnick, Jonathan Gordon, José Miguel Hernández-Lobato:
Predictive Complexity Priors. AISTATS 2021: 694-702

Yutian Chen, Abram L. Friesen, Feryal Behbahani, Arnaud Doucet, David Budden, Matthew Hoffman, Nando de Freitas:
Modular Meta-Learning with Shrinkage. NeurIPS 2020

Jonathan Gordon, John Bronskill, Matthias Bauer, Sebastian Nowozin, Richard E. Turner:
Meta-Learning Probabilistic Inference for Prediction. ICLR (Poster) 2019

**Summary Of The Paper:**

The paper proposes to use neural networks with so-called "stochastic local winner-takes-all (LWTA)" activations in the place of standard ReLU activations. These stochastic LWTA activations result in a model with sparse representations. The sparse gating is treated as a random variable. The authors also treat the weights of the neural network as random variables. Both sets of random variables are learnt using variational inference.

**Summary Of The Review:**

While this paper does have some strengths (strong empirical results, great set of ablation studies), it is let down by many issues of clarity, a lack of novelty, and a somewhat narrow set of main experiments.

++++ Post-revision update ++++

The authors have addressed many of my concerns, including correcting some of my misunderstandings, improving the clarity of the text in many ways, and adding additional experimental results in the form of ablations, baselines, and a new task. With this in mind, I have increased my score from 3 to 5.

I have not increased my score further as I still find some parts of the presentation slightly confusing (e.g. I don't think the word "principals" is much better than the word "argument") and more importantly, I still believe that having experiments only with image classification tasks is slightly too narrow.

---

> ### Author Response · Authors · 2021-11-19
> **Response to reviewer 3Mcm (part1)**
>
> $\textit{\textbf{Lack of novelty}}:$
> - In the revised version of our paper, we have rephrased the mentioned (admittedly) strong statement on novelty, and subsequently cited the suggested probabilistic approaches to ML, including BMAML [6], PredCP [15], Chen [16]; this is inline with the suggestions of other reviewers as well. We also underline that we provide comparisons to these methods, as we will explain in the following.
>
> $\textit{\textbf{Somewhat narrow set of main experiments}}:$
>
> (i) We have added new comparisons (Table 1) considering the CIFAR-100 5-way image classification dataset, and a multitude of alternative related approaches, including the methods the reviewers proposed.
>
> (ii) Unfortunately, our available computational resources could not afford evaluating on ShapeNet View Reconstruction task as used in Gordon [17].
>
> (iii) Tables 2 (main paper) and B1 (supplementary) provide the requested ablation studies on the differences between deterministic and stochastic formulations of LWTA. In all cases, we have obtained a meaningful accuracy improvement.
>
> (iv) We have also added a revised Table 4, where we provide an ablation study on the effect of sample size $B$ at testing time, as requested. In addition, in supplementary, Section E, Fig. 5, we provide graphical illustrations of the effect of sample size, $B$, on accuracy.
>
> $\textit{\textbf{Lack of clarity}}:$
> 1. Throughout the text, "argument" means a mathematical construct. In the revised paper, we have replaced the term "argument" with the term "principle" to facilitate the understanding.
> 2. We have revised the usage of word inference, and employed it only in the probabilistic sense. Thus, we used the word "prediction" to refer to test time prediction.
> 3. We have revised the notation for vectors $y$, and consistently use $y \in \mathbb{R}^{R \cdot J}$ as the output of a stochastic LWTA layer. This is composed of $R$ subvectors of $y_r \in \mathbb{R}^J$, where each of them consists of $J$ components $y_{r,j} \in \mathbb{R}$. Note also that, in a similar fashion, $\xi_r$ denotes the $r$-th subvector of $\xi$; it holds $\xi_r \in \mathrm{one}$_$\mathrm{hot}(J)$.
> 4. We emphasize that our approach does $\textit{not}$ pick the unit with maximum value. Instead, as Eqs. (6) and (7) show in the revised paper, $\textit{all}$ units in a block produce some output. This is in stark contrast with MoE layers, where the whole block generates a single scalar output. Turning to the process of output generation, each block selects a winner unit; that unit presents its original linear computation to the layer output; all the rest of the block units present zero output. We emphasize that winner selection is $\textit{not}$ deterministic; we sample the winner according to Eq. (7). It is true that the probabilities of selection stem from a softmax which can be similar to a gating function. However, we do not use the probabilities to compute a weighted sum, but to sample an one hot vector that eventually boils down to selecting a single winner. We stress that this process is input driven, similar to the gating function of MoE, contrary to the reviewers' comment. We hope that these clarifications combined with the paper revisions will facilitate understanding.

---

> ### Author Response · Authors · 2021-11-19
> **Response to reviewer 3Mcm (part2)**
>
> 5) The motivation behind this work comprises the following concepts: (i) sparse representations, as a product of the fact that only one unit per LWTA block presents a non-zero output to the next layer, but all units do present outputs to the next layer; and (ii) representations which are stochastic, in the sense that each time the same input is presented to the layer, a different layer output may be produced, as winner sampling may pick a different winner. These two core properties of the proposed layer outputs (learned representations) are substantially different from mainstream approaches. As the efficacy of ML strongly relies on how well the learned representations generalize across tasks, we can claim that the reported improvements in empirical performance are due to the quality/potency of these representations. However, in the revised paper we replace the term "potency" with "generalization capacity".
> 6) We have changed it to "model-agnostic".
> 7) We have revised the notation to resolve the apparent confusion; please see second paragraph of Section 3.1.
> 8) In the revised paper, Section 2 includes related work.
>
> (9,10,11,12,14,15,16,17,19). We resolved that issues.
>
> 13) We agree with this comment, and in fact the paper does adopt a very principled stochastic construction: the Categorical distribution that we sample the winners from represents a valid method of imposing a distribution over the winners; similarly, the imposition of a spherical Gaussian prior over the weights and derivation of the corresponding variational posterior is principled and appropriate.
> 18. We provide the requested equation in (14).
> 19. Exactly, the terms "neuron" and "unit" were used interchangeably. However, in the revised paper we only use the term "unit" for consistency.
> 20) Annealing the outer step size to 0 has been also used in [18], and shown to improve optimization and generalization.
> 21) As mentioned in the supplementary material, we follow the same architecture for the deep neural network as the one used by [19]. For Omniglot, the network is composed of 4 convolutional layers with 64 filters, 3 x 3 convolutions and 2 x 2 strides, followed by a Batch Normalization layer and the final values of each layer are processed by an activation function. For both Mini-Imagenet and CIFAR-100, we use 4 convolutional layers with 32 filters to reduce overfitting like [20], 3 x 3 convolutions followed by  Batch Normalization layer and 2 × 2 max-pooling layer, with the values of each layer finally passed again through an activation function (ReLU or LWTA; in the latter case, units are grouped into blocks).
>
> (22,23). All results have been updated with more comparisons and a fresh and rectified presentation. Please check updated Section 4.
>
> 24) The included parameters of each baseline for the outcomes of Table 5 are:
> - PLATIPUS [8]: $\Theta = ${$\mu_{\theta},\sigma^2_{\theta},v_{\theta}, \gamma_p, \gamma_q$}
> - BMAML [6]: $\Theta=[\theta^m]_{m=1}^5$, for using 5 particles
> - ABML [9]: $\theta=${$\mu_{\theta},\sigma^2_{\theta}$}
> - MAML [21]: $\theta=${$\mu_{\theta}$}
> - FOMAML [21]: $\theta=${$\mu_{\theta}$}
> - Reptile [22]: $\theta=${$\mu_{\theta}$}
> - StochLWTA-ML: $\theta=${$\mu_{\theta},\sigma^2_{\theta}$}
>
> (25,26). In the revised paper, we have replaced Fig. 2(b) with a figure which includes comparisons to the probabilistic ML models ABML, BMAML, PLATIPUS.
>
> $\textit{\textbf{Other comments}}:$
> 1. See point 4 above. See Section 2.1 in the revised paper.
>
> (2,3). Resolved
>
> 4. We have revamped Section 4. See reply to (22,23) above.

---

### Official Review · Reviewer_czsy · 2021-11-03

**Correctness:** 4
**Technical Novelty And Significance:** 2
**Empirical Novelty And Significance:** 2
**Recommendation:** 5
**Confidence:** 4

**Main Review:**

[Strengths]

* The idea to embed data-driven/learned sparsity in a meta-learning framework is really interesting. These methods have been shown to improve robustness, are open to continual learning and have been shown to enable learning independent mechanisms. It is a nice idea and this is backed up by competitive performance of the algorithm compared to original instantiations of MAML.

* The paper is well written and clear. As the reviewer of this paper, I feel that I have sufficient information and understanding to implement it myself

[Weaknesses]

* I find the methodological advances proposed in the manuscript too incremental for publication at ICLR. The SLWTA method proposed within appears to be the method of Panousis et al. (2021) (cited in text) but adapted to the MAML paradigm.

* The probabilistic model introduced by the authors bares many simiilarities with other data-driven sparsity models that are inline with Panousis et al. (2021). There is:
1. https://arxiv.org/abs/1805.10896 - Adaptive Network Sparsification with Dependent Variational Beta-Bernoulli Dropout
2. https://arxiv.org/abs/1912.02290 - Hierarchical Indian Buffet Neural Networks for Bayesian Continual Learning
Both who learn modularity with a similar ELBO as proposed by the authors.

* I think there are too few competiting baselines. Why were more recent gradient-based ML algorithms not considered such as Bayesian MAML (https://arxiv.org/abs/1806.03836) which first framed MAML in a Bayesian perspective?




**Summary Of The Paper:**

This paper proposes a stochastic local winner-takes-all (SLWTA) approach to learn data-driven (stochastic) sparsity. This is motivated by the limited availability of training data, especiially in few-shot meta learning scenarios. The authors propose a meta-learning algorithm for their SLWTA approach. As this is a probabilistic approach, one can draw from the approximate posterior at inference to obtained the posterior predictive distribution. The authors compare their method against MAML, FOMAML and Reptile on the Omniglot and Mini-Imagenet displaying better performance than the origiinal MAML algorithm.

**Summary Of The Review:**

I like the idea to embed an stochastic LWTA approach into meta-learning. It makes sense and the motivations are well presented. However, the technical novelty of the paper is lacking and does not offer enough methodological advancement to be a strong candidate for acceptance.

**** Post-rebuttal ****
I have decided to increase my score from 3 to 5. My reasoning is in my response to the author rebuttal.

---

> ### Author Response · Authors · 2021-11-19
> **Response to reviewer czsy**
>
> 1) The SLWTA method proposed within appears to be the method of Panousis et al. (2021) (cited in text) but adapted to the MAML paradigm.
> $\textbf{\textit{Answer}}:$
> - It is true that the proposed approach shares similar foundational principles with Panousis et al. (2021). However, there also are major differences in how these principles are made use of, since adversarial training poses different challenges and requirements compared to Meta-Learning (ML). For instance, Panousis et al. (2021) employs stochastic sampling of the synapses and/or convolutional kernels; this is performed by postulating extra "utility" indicator latent variables over the network synapses and/or convolutional filters, and imposing Indian Buffet Process (IBP) priors over them. Such an "architecture-wise" stochastic (sampling) process has been of relevance to the adversarial robustness setting, but it has been of no usefulness to ML. On the other hand, Panousis et al. (2021) obtain point-estimates for the network weights; in contrast, we have obtained much better outcomes in the context of ML by fitting (variational) posteriors over them; it seems that stochastic network weights facilitate generalization in the context of ML.
> - To summarize, our paper does share a common core methodological aspect with Panousis et al. (2021), but it also has major methodological differences. The fact that we share some common foundational principle does not severely limit novelty; if that was the case, for instance, no paper proposing variants of the convolutional layer would have ever been published in ICLR.
>
> 2) The probabilistic model introduced by the authors bares many similarities with other data-driven sparsity models that are inline with Panousis et al. (2021). There is:
> i) https://arxiv.org/abs/1805.10896 - Adaptive Network Sparsification with Dependent Variational Beta-Bernoulli Dropout
> ii) https://arxiv.org/abs/1912.02290 - Hierarchical Indian Buffet Neural Networks for Bayesian Continual Learning Both who learn modularity with a similar ELBO as proposed by the authors.
> $\textbf{\textit{Answer}}:$
> - We stress that these approaches use some methodological principles related with our SLWTA units; however, these are not the same, and have not been evaluated in the context of ML. Thus, we cannot provide comparisons to these as well, even though we would love to.
>
> 3) I think there are too few competing baselines. Why were more recent gradient-based ML algorithms not considered such as Bayesian MAML (https://arxiv.org/abs/1806.03836) which first framed MAML in a Bayesian perspective?
> $\textbf{\textit{Answer}}:$
> - In the revised version of our paper, we add plenty of competitive approaches, including the reviewer-requested Bayesian MAML (BMAML) [6], as well as LLAMA [7], PLATIPUS [8], and ABML [9]. As we show in the revised Table 1, our approach yields the highest predictive performance in all benchmarks, with statistically significant differences from the second-based approaches in all cases.
> - In addition, in the revised Table 5 we emphasize that this accuracy improvement comes with a training time improvement that averages at 77% compared to the aforementioned time-consuming models ABML, BMAML and PLATIPUS.

---

> > ### Comment · Reviewer_czsy · 2021-11-29
> > **Thanks a lot for your rebuttal and for updating the manuscript!**
> >
> > Thanks a lot for the response. I appreciate the revised manuscript with the extra experiments.
> >
> > However, I disagree with the authors on the novelty of the work. It is claimed in the manuscript that
> > >For the first time in the literature of deep network-driven ML, we employ a mechanism that inherently learns to extract sparse
> > data representations. This consists in replacing standard unit nonlinearities (e.g., ReLU) with a unit competition mechanism.
> >
> > The work on Recurrent Independent Mechanisms [1] proposed this recently by using self-attention as a competition mechanism to determine which modules should be activated in an RNN and thus learned sparsity in the network. Further, the concept of learning sparse data representations has been approached in various pieces of work such as the ones cited in my review but also in other fields such as multi-task learning where the task-sparsity is governed by Categorical distributions [2].
> >
> > I reiterate that I like the idea of this paper - namely learning modularity/sparsity through meta-learning. I appreciate the extra experiments and updated manuscript and therefore increase my score from 3 to 5. However, I still find the novelty lacking and thus cannot overly push for acceptance and would constitute myself as borderline.
> >
> > [1] https://arxiv.org/abs/1909.10893
> > [2] https://arxiv.org/abs/1908.09597
> >
> > Additionally:
> >
> > * I cannot find any motivation in the manuscript for why learned modularity/sparsity through meta-learning is advantegeous. As an interested reader, I would like to understand why this is necessary for few-shot learning
> >
> > * I would also appreciate a set of experiments on standard supervised tasks to appreciate whether the SLWTA method proposed improves on tasks.

---

### Author Response · Authors · 2021-11-19
**References**

The references used for reviewers' responses are:

[1] Lansner, A. Associative memory models: from the cell-assembly theory to biophysically detailed cortex simulations. Trends in Neurosciences, 32(3), 2009.

[2] Carlos Riquelme, Joan Puigcerver, Basil Mustafa, Maxim Neumann, Rodolphe Jenatton, André Susano Pinto, Daniel Keysers, Neil Houlsby: Scaling Vision with Sparse Mixture of Experts. CoRR abs/2106.05974 (2021)

[3] Juho Lee, Saehoon Kim, Jaehong Yoon, Hae Beom Lee, Eunho Yang, Sung Ju Hwang, Adaptive Network Sparsification with Dependent Variational Beta-Bernoulli Dropout, arXiv preprint arXiv:1805.10896, 2018

[4] Samuel Kessler, Vu Nguyen, Stefan Zohren, Stephen Roberts, Hierarchical Indian Buffet Neural Networks for Bayesian Continual Learning, UAI 2021.

[5] Dmytro Mishkin, Nikolay Sergievskiy, Jiri Matas, Systematic evaluation of CNN advances on the ImageNet, CVIU 2016

[6] Yoon, J. and Kim, T. and Dia, O. and Kim, O. and Bengio, Y. and Ahn, S., Bayesian model-agnostic meta-learning, In Neural Information Processing Systems, 2018

[7] Erin Grant, Chelsea Finn, Sergey Levine, Trevor Darrell, and Thomas Griffiths. Recasting gradient-based meta-learning as hierarchical bayes, In International Conference on Learning Representations, 2018.

[8] Chelsea Finn, Kelvin Xu, and Sergey Levine. Probabilistic model-agnostic meta-learning. In NeuralInformation Processing Systems, 2018.

[9] Sachin Ravi and Alex Beatson.  Amortized bayesian meta-learning.  InInternational Conference onLearning Representations, 2019.

[10] Rupesh Kumar Srivastava, Jonathan Masci, Faustino Gomez, Jürgen Schmidhuber, Understanding Locally Competitive Networks, In International Conference on Learning Representations, 2015.

[11] Tiago M. Fragoso and Francisco Louzada Neto, Bayesian model averaging: A systematic review and conceptual classification, Statistical Science, 2015

[12] Charles Blundell, Julien Cornebise, Koray Kavukcuoglu and Daan Wierstra, Weight Uncertainty in Neural Networks, In Neural Information Processing Systems, 2015

[13] Eric Jang, Shixiang Gu, and Ben Poole. Categorical reparameterization with gumbel-softmax. In International Conference on Learning Representations, 2017

[14] Chris J. Maddison, Andriy Mnih, and Yee Whye Teh. The concrete distribution: A continuous relaxation of discrete random variables. In International Conference on Learning Representations, 2017.

[15] Eric Nalisnick , Jonathan Gordon, José Miguel Hernández-Lobato, Predictive Complexity Priors, In Neural Information Processing Systems, 2021.

[16] Yutian Chen, Abram L. Friesen, Feryal Behbahani, Arnaud Doucet, David Budden, Matthew Hoffman, Nando de Freitas, Modular Meta-Learning with Shrinkage, In Neural Information Processing Systems, 2020

[17] Jonathan Gordon, John Bronskill, Matthias Bauer, Sebastian Nowozin, Richard E. Turner: Meta-Learning Probabilistic Inference for Prediction. ICLR (Poster) 2019

[18] Ilya Loshchilov, Frank Hutter, SGDR: Stochastic Gradient Descent with Warm Restarts, In International Conference on Learning Representations, 2017

[19] Oriol Vinyals, Charles Blundell, Timothy Lillicrap, Koray Kavukcuoglu, Daan Wierstra, Matching Networks for One Shot Learning, In Neural Information Processing Systems, 2016.

[20] Sachin Ravi and Hugo Larochelle, Optimization as a Model for Few-shot Learning, In International Conference on Learning Representations, 2017

[21] Chelsea Finn, Pieter Abbeel, and Sergey Levine. Model-agnostic meta-learning for fast adaptation of deep networks. InInternational Conference on Machine Learning, 2017

[22] Alex Nichol, Joshua Achiam, John Schulman, On First-Order Meta-Learning Algorithms, URL https://arxiv.org/abs/1803.02999, 2018

[23] Noam Shazeer, Azalia Mirhoseini, Krzysztof Maziarz, Andy Davis, Quoc V. Le, Geoffrey E. Hinton, Jeff Dean: Outrageously Large Neural Networks: The Sparsely-Gated Mixture-of-Experts Layer. ICLR (Poster) 2017

---

### Author Response · Authors · 2021-11-19
**ICLR 2022 rebuttal authors' response**

We have uploaded a revision of the paper, both the main manuscript as well as Supplementary material appended to the main paper (this is needed due to space limitations). Therein, we address all reviewers concerns.

- We updated our introduction and added "Related work" section, to better reflect how StochLWTA-ML differs from existing probabilistic and sparsity-based models. We have also included more (reviewer-requested) comparisons to existing methods (in Section 4).
- Since reviewers noticed an unjustifiedly slower training time of the First-Order ML  Reptile algorithm compared to the Second-Order MAML, we re-run all the experiments on a different GPU, which would afford a task batch size of 50; this rectified the observed artifact. More details are provided in our reviews' replies.
- We have rewritten the description of our proposed approach in Section 3 to facilitate the understanding of our training and prediction processes.
- Additional experiments on CIFAR-100 5-way benchmark can be found in most tables of Section 4.
- In the main paper's Section 4.3.2, we included an ablation study regarding the effect of block size $J$.
- In Table 5, we added the number of training iterations that each of the baseline methods needed to converge.
- We replaced Fig. 2(b) with a convergence figure that compares StochLWTA-ML to other evaluated probabilistic approaches, namely ABML [9], BMAML [6] and PLATIPUS [8]; for brevity, we focused on the Omniglot 20-way 1-shot benchmark.
- In Tables 2 (main paper) and B1 (supplementary), we provided the requested ablation studies, comparing the performance of stochastic and deterministic LWTA configurations; deterministic LWTAs select the winner as the unit with the $\textit{highest}$ linear computation within a block.
- In Section D of the Supplementary, we illustrate the performance of our model on the Mini-Imagenet 5-way 1-shot setting with different task batch sizes, in order to justify the choice of task batch size of 50 for both training and testing.
- In a similar vein, Section E of the Supplementary shows our model's predictive performance on different sample sizes $B=${$1,\dots. ,10$} at testing time.
- In Section F of the Supplementary, we analyze the number of trained parameters of each evaluated baseline, as well as of the proposed approach.

---

> ### Comment · Reviewer_3Mcm · 2021-11-26
> **Increased my score**
>
> The authors have addressed many of my concerns, including correcting some of my misunderstandings, improving the clarity of the text in many ways, and adding additional experimental results in the form of ablations, baselines, and a new task. With this in mind, I have increased my score from 3 to 5.
>
> I have not increased my score further as I still find some parts of the presentation slightly confusing (e.g. I don't think the word "principals" is much better than the word "argument") and more importantly, I still believe that having experiments only with image classification tasks is slightly too narrow.

---

> > ### Author Response · Authors · 2021-11-27
> > **Follow up response to reviewer 3Mcm**
> >
> > - In the revised paper, we have used the term $\boldsymbol{\textrm{principles}}$ and not $\textrm{principals}$. Could the reviewer please let us know what are the concerns over the usage of this term?
> >
> > - We have extended Section 4 and Supplementary material with further experiments, as well as the requested ablation studies. We have evaluated our proposed approach and baselines on standard few-shot learning benchmarks, as the majority of our competitors did.  We believe it is unfair to be judged for this, since our available computational resources could not afford evaluating on ShapeNet View Reconstruction task as used in Gordon [17].

---

> > > ### Comment · Reviewer_3Mcm · 2021-11-28
> > > **Clarification**
> > >
> > > Regarding clarity. The choice of the word "principle" is just an example, I simply find that the meaning is not clear in the context of the paper. However, I am afraid that I find the overall presentation of the work hard to grasp due to many factors including word choice, organization of ideas, and motivation. I encourage the authors to solicit feedback from a range of readers at different levels of machine learning expertise and in different sub-fields.
> > >
> > > Regarding the experimental evaluation. I appreciate the amount of work that went into all of the additional experimental evaluations, including ablation studies and additional baselines. This is reflected in my increased score. However, I believe that it is a flaw of this work that all of the experimental evaluation was conducted in the image classification setting. The ShapeNet View Reconstriction task was simply a suggestion. The authors would have been welcome to use a different task that required a smaller computational budget.

---

> ### Author Response · Authors · 2021-11-27
> **Follow up rebuttal authors' response**
>
> We believe we have addressed all the issues raised by the reviewers. We would like to respectfully ask the reviewers to reconsider their scores. We are also happy to address any further questions.

---

### Decision · Program_Chairs · 2022-01-20

**Decision:**

Reject

**Comment:**

This paper presents a meta-learning method where the standard ReLU activation units are replaced by the stochastic LWTA units to learn data-driven sparse representation.  Most of reviewers like the idea of embedding the stochastic LWTA into a MAML framework. Initial assessment pointed out the lack of clarity in various places in the paper. Authors did a good job in the author’s rebuttal period, to clarify the paper. Experiments demonstrated the competitive performance over existing meta-learning methods. Two of reviewers raised their overall score to 5 (from 3). However, all reviewers have concerns in the incremental technical novelty and feel that presentation should be improved to make the paper more clear and friendly to readers. While the idea is interesting, which is backed up by experiments, the paper is not ready for the publication at the current stage. I encourage to resubmit the paper after addressing these concerns.